# Integration of serial self-testing for COVID-19 as part of contact tracing in the Brazilian public health system: A pragmatic trial protocol

**Rebecca K. Green** [1]*, **Camilo Manchola**[2], **Emily Gerth-Guyette**[1], **Michelle Oliveira Silva**[3], **Raissa Stephanie**[3], **Tainá dos Santos Soares** [4], **Luiza Bastos Gottin**[2], **Milena Coelho**[2], **Kimberly E. Green**[1], **Alexandre Dias Tavares Costa** [4], **Dhélio Batista Pereira** [3]

**1** PATH, Seattle, Washington, United States of America, **2** Global Health Strategies, Rio de Janeiro, Rio de Janeiro, Brazil, **3** Centro de Pesquisa em Medicina Tropical de Rondônia (CEPEM), Porto Velho, Rondônia, Brazil, **4** Laboratory for Applied Science and Technology in Health, Carlos Chagas Institute, Fiocruz Paraná, Curitiba, Paraná, Brazil

* rgreen@path.org

**Data Availability Statement:** Deidentified research data will be made publicly available when the study is completed and published.

## Abstract

The coronavirus disease (COVID-19) pandemic has led to an unprecedented public health crisis. Insufficient testing continues to limit the effectiveness of the global response to the COVID-19 pandemic. Molecular testing methods such as reverse transcriptase polymerase chain reaction (RT-PCR) continue to be highly centralized and are a sub-optimal option for population surveillance. Rapid antigen tests (Ag-RDTs) offer multiple benefits including low costs, high flexibility to conduct tests in a wide variety of settings, and faster return of results. Self-test Ag-RDTs (STs) have gained approval in several markets and offer the possibility to expand testing, reaching at-risk populations. While STs have the potential to assist the COVID-19 response, test result integrity, reporting, and appropriate linkage to care continue to hinder the widespread implementation of self-testing programs. This protocol presents a mixed-methods pragmatic trial (ISRCTN91602092) to better understand the feasibility of self-testing as part of a contact tracing strategy within the Brazilian public health system. Approximately 604 close contacts of 150 index cases testing positive for COVID-19 will be enrolled. Index cases will be randomized for their close contacts to participate in either serial (daily) self-testing over a 10-day follow-up period or a more traditional approach to contact tracing with a professional Ag-RDT at one time point post-exposure. Usability workshops and focus group discussions will also be conducted. This study protocol presents a comprehensive plan to assess the effectiveness, operational feasibility, and stakeholder preferences of a serial self-testing strategy for contact tracing within the Brazilian public health system. Our results will contribute to better understanding of the feasibility of a self-testing strategy within the public sector. Potential risks and limitations are discussed. Our findings will have important implications as governments continue working to mitigate the impact of COVID-19, particularly in the context of where to direct limited resources for testing and healthcare infrastructure.

**Registration:** This trial is registered at ISCTRN (ISRCTN91602092).

**Funding:** This project is funded by Unitaid (unitaid. org) in collaboration with Population Services International (psi.org) (Award #: 2017-16-PSI-STAR); funding was acquired by KEG. The funders had no role in study design, data collection and analysis, decision to publish, or preparation of the manuscript. Grant support for the electronic data capture system accessed through a standing collaboration between PATH and the University of Washington is from the National Center for Advancing Translational Sciences of the National Institutes of Health under Award Number UL1 TR002319.

**Competing interests:** The authors have declared that no competing interests exist.

# Introduction

The coronavirus disease (COVID-19) pandemic led to an unprecedented public health crisis. Insufficient testing continues to limit the effectiveness of the local and global response to the pandemic. Isolation and quarantine guidelines continue to evolve yet primarily rely on case identification and subsequent behavior modification for infected or exposed individuals. In many settings, the gold standard for COVID-19 diagnosis is reverse transcription polymerase chain reaction (RT-PCR), which is plagued by sparse availability of supplies, higher cost, slow turnaround times, and its highly centralized nature [1]. These challenges make RT-PCR difficult to deploy widely and therefore not an optimal candidate test as a public health tool for population surveillance and effectively interrupting transmission chains [2].

Rapid antigen tests (Ag-RDTs) for professional and self-test use offer multiple benefits in comparison to RT-PCR, including low costs and increased portability. Ag-RDTs can expand access to COVID-19 testing in places that do not have molecular testing capacity and results can be returned quickly, facilitating faster reporting and subsequent linkage to care. WHO recommends use of rapid antigen tests and self-tests for kits meeting minimum performance requirements in priority use cases [3–6]. Further, rapid antigen tests may be more suitable in settings where people have been previously infected and molecular testing methods continue to return positive results due to residual viral fragments.

## Self-testing

Successful strategies to broaden access to testing include the use of self-tests (STs) [7–9]. Self-testing regimens for COVID-19 are a promising method to identify infectious individuals, interrupt transmission chains, and reduce demand on health facilities while addressing many of the usual barriers to uptake of services [2, 10, 11]. In addition to enabling more timely isolation to minimize onward transmission, swift diagnosis can also prompt clinical intervention, which may improve individual patient prognosis, particularly given the availability of new antivirals. Self-testing has shown high levels of acceptability, with many countries implementing large-scale programs to access at-home tests for free, and can increase equity by providing more testing options [7, 12]. In short, there is evidence that self-testing for COVID-19 is feasible and acceptable, with both national and global recommendations to use self-tests and some specific products receiving emergency use authorization from the US Food and Drug Administration (FDA) and WHO [13, 14].

While self-testing has the potential to contribute significantly to the COVID-19 response, it also comes with limitations. Firstly, available antigen tests may have variable performance in asymptomatic individuals [13, 15]. False negative results may prompt infected individuals to stop self-isolation and thereby contribute to virus transmission, while false positive results may lead to unnecessary stress, anxiety, and absences from work, school, and social activities. Secondly, self-testing results may not be reported and therefore missed by local and national health authorities. However, WHO guidelines describe the overall benefits of self-testing as outweighing these limitations. Clear communication on actions for positive and negative results, relevant support tools, efficient links to post-test counselling and easy access to results reporting are needed as key components of self-testing programs. The effectiveness of these approaches have already been well-established in HIV self-testing programs [16].

## Contact tracing

Early on in the COVID-19 pandemic, contact tracing was used to limit onward transmission and link at-risk individuals to testing and care [17]. Since then, both observational and modelling studies have shown that contact tracing is associated with better control of COVID-19 and

is growing increasingly important for today's surveillance strategies to guide outbreak response. The impact of contact tracing is mediated by a number of factors, including the time it takes to identify and notify contacts and the number of positive cases that participate in contact tracing [18]. During periods of peak transmission, contact tracing efforts may be slowed, stymied, or abandoned all together when the number of cases exceeds the public health system's capacity to identify and follow-up with exposed cases, as experienced during the omicron wave [19, 20]. In these cases, a self-testing regimen for exposed individuals may help interrupt transmission and control the outbreak.

In the context of contact tracing, self-testing facilitates even further decentralization of testing and allows for faster identification of infectious contacts, reaching at-risk populations, and generally mitigating unequal access to testing [15, 21]. However, to ensure equitable access to self-testing, the use of these steps must be integrated into public health system programs and strategies rather than simply making self-tests available as a consumer product. This is particularly true in places where self-tests are difficult to obtain, either logistically or financially. Serial self-testing may also be advantageous to contact-tracing efforts, as it allows exposed individuals to monitor themselves over time and does not rely on a single time point to determine infection status, particularly when individuals may not seek care during the recommended post-exposure period and the recommended on-label testing algorithm for many self-tests calls for testing twice in the event of a negative test. Serial self-testing as part of a public health system contact tracing strategy may be a viable option to avoid multiple follow-up visits and allow both patients and healthcare providers to benefit from the decentralized and flexible nature of self-testing.

## Objectives and hypothesis

The primary aim of this study is to evaluate the effectiveness of contact tracing supported by serial self-testing (testing daily for up to 10 days) among exposed individuals compared to routine contact tracing at one time point. This study also aims to evaluate the operational feasibility of self-testing for contact tracing within the Brazilian public health system, explore the barriers and facilitators at the provider and patient levels that mediate use of COVID-19 self-tests, and assess adherence to quarantine, isolation, and treatment guidelines. This study hypothesizes that serial self-testing of primary close contacts will identify more positive cases than routine contact tracing at a single timepoint post-exposure in a facility-based health care setting.

## Materials and methods

### Design and setting

This is a mixed-methods, two-arm randomized pragmatic trial within the public health system of two municipalities in Brazil. The study will be conducted at the Centro de Pesquisa em Medicina Tropical de Rondônia (CEPEM) in Porto Velho, Rondônia and several health units in Curitiba, Paraná. The health system within each municipality is structured through localized health units that are responsible for providing care to a specific catchment area. Health units are generally staffed with nurses, technicians, community health workers, doctors, and pharmacists. Professional antigen testing for COVID-19 is widespread within the public health system and pharmacies, though RT-PCR testing remains the laboratory test of choice for patients in the acute phase with moderate to severe symptoms [22]. Patients with no or mild symptoms may not receive a confirmatory PCR test and are typically advised to isolate [23]. Contact tracing practices conducted by the public health system have varied between study sites based on

health system capacity, COVID epidemiology, and caseload, though individuals have generally been instructed to notify their close contacts.

## Population

Patients aged 7 or older testing positive for COVID-19 at any participating health unit are eligible to be enrolled into the study as index cases. Approximately 604 close contacts of 150 index cases testing positive for COVID-19 per local standard of care testing practices will be enrolled. All index cases will complete a contact elicitation interview to identify close contacts who have been exposed to SARS-CoV-2 Close contacts will be invited to participate in the study following exposure notification. Close contacts of the index case will be eligible for enrollment as primary close contacts if they are 7 years of age or older and have been exposed to an enrolled index case within 2 days of index case symptom onset or within 7 days of index case positive test result. For the purposes of this study, "exposure" is defined as being within 1 meter of the index case for more than 15 minutes or having physical contact without appropriate personal protective equipment.

## Intervention

Index cases will be randomized such that their close contacts are assigned to either control or intervention. Randomization will be performed 1:1 at the index case level such that all immediate close contacts of an index case are randomized to the same arm. This is the most practical way to randomize, as at least some primary close contacts are expected to share a household with the index case and this approach will minimize contamination between the arms. The data manager will create the allocation sequence through computer-generated random numbers and will store this information in a locked Excel file. Only the data manager will have access to the full sequence. Study staff will reveal participant assignment upon enrollment by accessing a limited version of the file.

Primary close contacts enrolled in the intervention arm will complete an enrollment visit in person, either at the health unit or at home. Following consent, participants will complete a baseline questionnaire, a supervised self-test and a health worker will independently perform a rapid test. The order in which these tests are conducted will be determined based on study ID and balanced within the arm to avoid test result bias related to sample depletion. Participants will then be provided with 10 self-tests of the same brand and lot to perform daily over the subsequent 10 days and will be contacted daily to complete a brief questionnaire.

The study will also provide additional tests to household members of primary close contacts enrolled in the intervention arm and invite them to submit anonymous data around their use of the tests. These household members will not be consented or enrolled into the study and will receive no follow-up from the study team.

Primary close contacts enrolled in the control arm will complete an enrollment visit either at the health unit or at home. Following consent, participants will complete a baseline questionnaire and a health worker will perform a rapid test. Participants will be contacted daily over the following 10 days to complete a brief study questionnaire. Fig 1 details the schedule of enrollment for each participant group and Fig 2 illustrates the overall study design.

## Outcomes

The primary outcome will be the proportion of primary exposure close contacts who test positive for COVID-19 per index case in the intervention arm compared to the control arm. This will be captured through the participant follow-up surveys. Subjects with missing outcome data will be excluded from analysis. For the purposes of analysis, participants in the

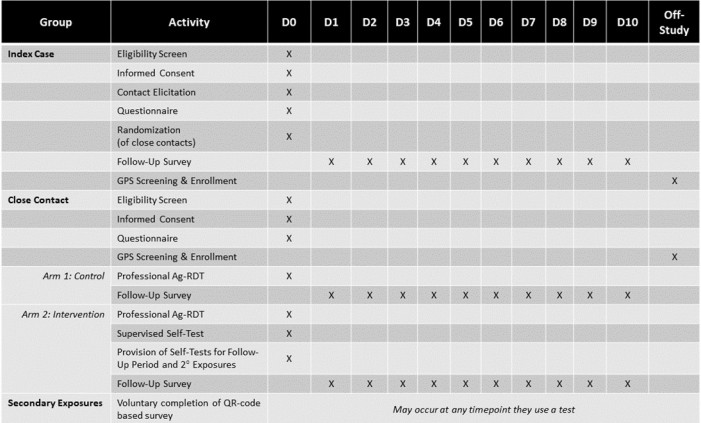

**Fig 1. Schedule of enrollment, interventions, and assessments.** A schedule of data collection activities by participant group.

intervention arm will be considered to have complete primary endpoint data if at least three test results (consecutive or otherwise) are submitted over the 10-day follow-up period. For participants in the control arm, a professional Ag-RDT result available at the enrollment visit will be considered complete.

Additional endpoints captured in the follow-up survey include proportion of close contacts testing positive who report adhering to recommended isolation guidelines and proportion of exposed contacts who report test results per local guidelines (intervention arm only). During the enrollment visit for participants enrolled in the intervention, study staff will document whether the participant correctly performs the self-test (per the manufacturer's instructions for use) and the result of both self-test and professional rapid test to assess concordance between the two. Diagnostic accuracy assessment against a gold standard test is out of scope for this study design.

Acceptability, preferences, and user perspectives on self-testing will be assessed across stakeholders, including study participants and health professionals, through focus group discussions; insights into user needs for instructions and training will be captured through user workshops.

## Statistical considerations

**Sample size.** This study is powered to detect a 7.5% difference in positive cases identified between the intervention and control arm. This threshold was chosen through discussions with stakeholders to balance detection of a clinically meaningful outcome with an understanding of the likely rates of infection in a highly vaccinated population. Prior work at this site with COVID-19 testing of close contacts yielded a 30% PCR positivity rate among close contacts during a period of low to moderate transmission. To account for increases in vaccination coverage, high rates of prior infection, and the likelihood of low transmission following the most recent wave, this population is estimated to have a 20% test positivity rate. Based on the established performance characteristics of Ag-RDTs, serial self-testing is estimated to identify up to 75% of those cases. Using Eq 1, where $z_{\alpha/2} = 1.96$ and $z_\beta = 0.842$, a total of 550 participants are needed (275 per Arm). To account for attrition in longitudinal data and exclusion of unevaluable cases, we will increase the sample size estimate by 10% to enroll a total of 604 close contacts (302 per Arm). To achieve this, approximately 150 index cases are needed, with each

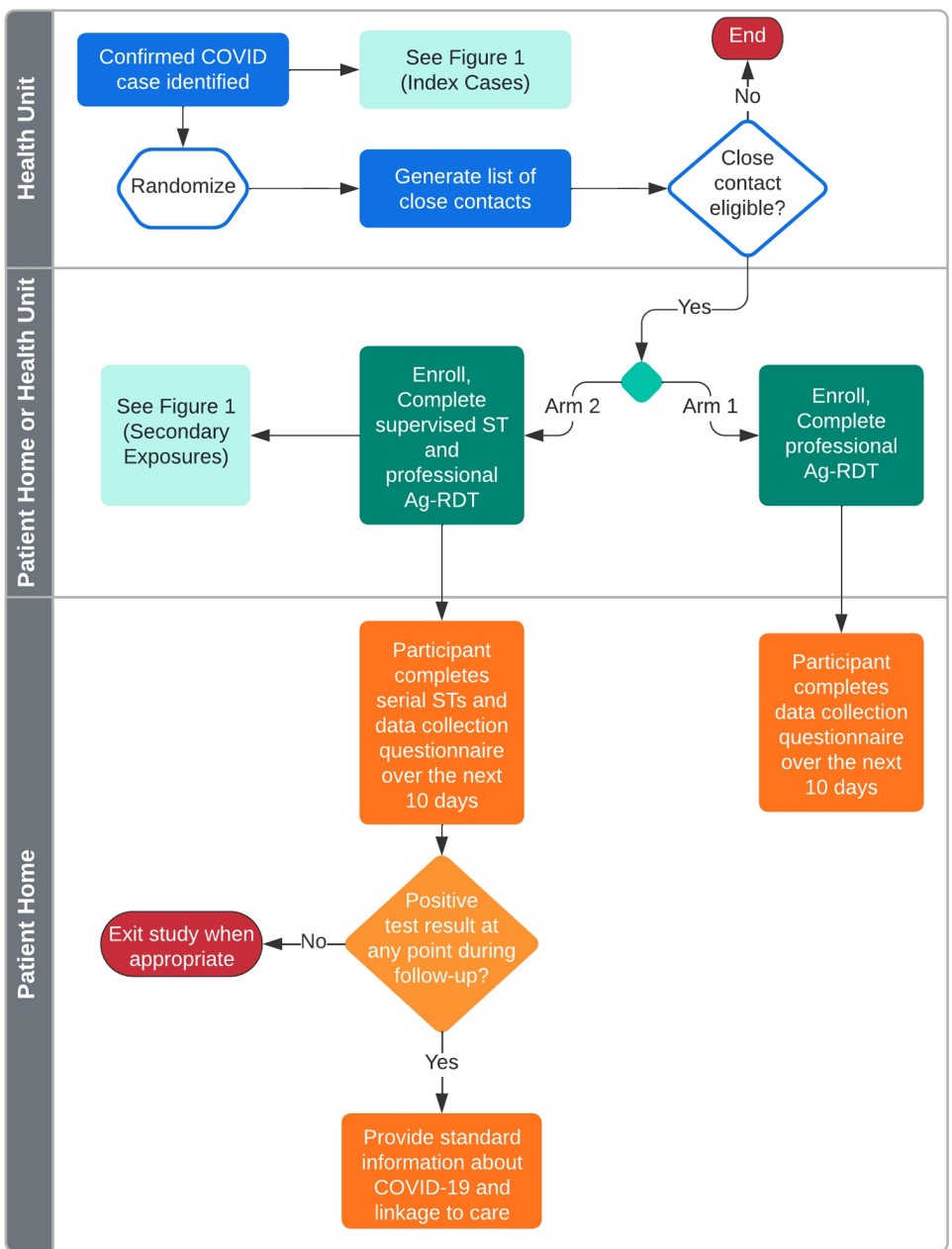

**Fig 2. Study design.** A schematic of the study design.

index case leading to an estimated average of 4 close contacts (75 per Arm). These estimations are based on prior work at the study site [24].

$$N = \frac{2\left(z_{\frac{\alpha}{2}} + z_\beta\right)^2 * (p_0(1 - p_0)) + (p_1(1 - p_1))}{\Delta^2} \tag{1}$$

Sample size for the usability workshop will be 6–10 participants and sample size for focus groups will be up to 15 participants [25].

**Data collection and management.** All data will be entered by study staff and participants or their caregivers directly into a secure, electronic database (REDCap) managed by the University of Washington Institute of Translational Health Sciences [25]. Participants may opt to complete the follow-up survey by phone with study staff or by WhatsApp message link. For participants opting to receive the daily WhatsApp message, a message containing the survey link will be automatically sent each morning. If they do not complete the survey by the evening, they will be sent an automated reminder message. An automated script will alert the study team of any data anomalies (e.g., duplicate responses) and participant needs for follow-up (e.g., participant requested follow-up or participant has missed two or more consecutive surveys). Structuring the follow-up data collection in this way minimizes staff burden and may provide insights into how a similar system could be set up to have the greatest likelihood of influencing health behavior and ensuring test results are appropriately reported.

Secondary exposures will have the option to submit anonymous data by scanning a QR code to complete the survey, which may provide additional insight into alternative result reporting modalities. Usability workshops and focus group discussions will be recorded to ensure all detailed information can be captured appropriately.

## Safety considerations

This study poses minimal risk to participant safety, as it does not involve any medical intervention and biological sampling is within acceptable ranges. Only research staff who have been trained in best practices for specimen collection and infection prevention will be involved in specimen collection. All self-tests used in the study are approved for commercial sale and use in Brazil. All records will be kept confidential at each site and the sponsor will not have access to any records that directly identify the research participants.

## Ethical considerations

**Risks.** Study procedures do not represent significant risks to the participants beyond those associated with a nasal swab, such as pain, discomfort, and nosebleed, which will be mitigated through user training on proper sample collection. All participants will be made aware that taking part in any study activity is voluntary. All study team members will adhere to institutional procedures for infection control and will have adequate personal protective equipment to minimize risks related to COVID-19 transmission.

All decisions regarding clinical care will be made through referral to the local public health system. The study team will review data in real-time to ensure that any information provided on clinical symptoms is referred appropriately per local public health guidelines.

**Benefits.** Participants in this study will have convenient access to COVID-19 testing following an exposure. Household members of these participants will also have access to free COVID-19 self-tests should they wish to use them. There is no direct benefit to the community, however there may be indirect benefits by identifying more positive cases, which could reduce the spread of COVID-19 in the community.

**Special considerations for enrolling minors.** The written consent/assent process for this study is tailored to three different age groups that may be enrolled, per Brazilian regulations. For the youngest age group assenting (7–11 years of age), the caregiver (parent/legal guardian giving written consent) will perform the self-test on the child. The other two age groups (12–14 and 15–17 years of age) will perform the self-test on themselves, though they may receive help from their caregiver as needed. This determination is consistent with what would reasonably be expected of these age groups and Brazilian research ethics regulations. Additionally,

the self-tests being used in this study are indicated for children 2 years and older with adult supervision, so all use during this study will be on-label.

Enrolling children does not sufficiently alter the risk-benefit ratio to warrant excluding children from this study. The only additional risk this study poses to children is that they may be more likely to use the test incorrectly, which will be mitigated by conducting the test with caregiver supervision. Children may also benefit from the findings of this research given their participation in school and social group activities and that they have been disproportionately affected by the pandemic due to school closures. This study will show how serial self-testing can be used in families and whether it can be operationalized to be effective. Therefore, it would be unethical to exclude children from this study given their potential to benefit from the findings and minimal additional risk posed to them.

### Study timeline

This study has been approved with written consent by The National Commission for Research Ethics (CONEP, Brazil's national Institutional Review Board [IRB], approval number 59179922.9.1001.0011), the local IRBs in Porto Velho and Curitiba, and the WHO Ethics Review Committee. The study began recruitment December 5, 2022, and will run for approximately six months.

### Study registration

This study is registered at ISCTRN (ISRCTN91602092).

## Discussion

### Impact

This novel approach to contact tracing attempts to increase equitable access to essential diagnostics in the face of the worst pandemic in recent history. This study will generate data around the operational feasibility and effectiveness of a serial self-testing strategy in the context of the Brazilian public health system. While providing patients with 10 self-tests to perform daily is likely impractical, we are hoping further sensitivity analyses may reveal more optimal testing strategies to inform appropriate resource allocation. The study ultimately aims to generate evidence to support health policy makers in Brazil to understand whether this is a feasible tool to incorporate into the unified health system to support public health contact tracing/outbreak response efforts, which would reduce barriers to self-test access and promote use of self-tests. Finally, this study will join a growing body of evidence being generated to find optimal self-testing algorithms as part of non-pharmaceutical interventions to mitigate onward community transmission of COVID-19 [10, 11]. Evidence from these studies will be critical to inform policy and public health practice around the use and merits of self-testing through an epidemiological lens [2]. This work will also help codify lessons learned to leverage for future pandemic preparedness.

### Limitations

Study activities will be impacted by changing public health policies and guidelines as well as the evolving epidemiology of COVID-19. Where possible, these will be monitored throughout the study and tracked both administratively and through participant surveys. The methods outlined in this protocol are designed to be flexible enough to adapt to the local COVID-19 situation as needed while maintaining sufficient scientific rigor to fulfill study objectives. Additionally, follow-up data may be biased due to both the nature of being self-reported as well as

observed. Data from secondary exposures may also be influenced by the fact their household member is participating in a research study. Finally, performing daily self-tests for 10 days is likely not a cost-effective method of contact tracing for COVID-19. This strategy was selected in the interest of generating a robust data set with the potential to conduct and inform additional analyses regarding cost effective testing strategies. Commodity costs will also be tracked to support these ancillary analyses.

## Dissemination, stakeholder, and participant engagement

Stakeholders from the local municipalities and the Ministry of Health have been engaged throughout protocol development to better understand the local health system perspective when building the study's objectives and methodology. These channels of communication will remain open throughout the conduct of the study to ensure continued engagement, and results will be shared back with these stakeholders. Research results will be shared locally, at the participating facilities and health units through debrief meetings and short reports. In addition to local results sharing, study findings will be disseminated through a variety of channels, including engagement with the World Health Organization, donors, and peer-reviewed publications. At the close of their participation, participants will also be sent information about where they can find final study results and be notified of dissemination outputs and events.

## Supporting information

**S1 File. Approved study protocol.**
(PDF)

**S2 File. Example consent form used in study.**
(PDF)

**S3 File. SPIRIT checklist.**
(DOC)

## Acknowledgments

The authors would like to acknowledge the Coordination of Respiratory Diseases from the Ministry of Health of Brazil and the Health Secretariats from Porto Velho and Curitiba Municipalities for their valuable contributions throughout the protocol development process. The authors also acknowledge Stephanie Zobrist and Gonzalo Domingo for their technical guidance in the development of this protocol and Krista Granger for her project coordination support.

## Author Contributions

**Conceptualization:** Rebecca K. Green, Camilo Manchola, Emily Gerth-Guyette, Michelle Oliveira Silva, Luiza Bastos Gottin, Milena Coelho, Alexandre Dias Tavares Costa, Dhélio Batista Pereira.

**Funding acquisition:** Kimberly E. Green.

**Investigation:** Michelle Oliveira Silva, Raissa Stephanie, Tainá dos Santos Soares, Alexandre Dias Tavares Costa, Dhélio Batista Pereira.

**Methodology:** Rebecca K. Green, Camilo Manchola, Emily Gerth-Guyette, Michelle Oliveira Silva, Luiza Bastos Gottin, Milena Coelho, Alexandre Dias Tavares Costa, Dhélio Batista Pereira.

**Project administration:** Camilo Manchola, Emily Gerth-Guyette, Alexandre Dias Tavares Costa, Dhélio Batista Pereira.

**Resources:** Kimberly E. Green.

**Supervision:** Emily Gerth-Guyette, Kimberly E. Green, Alexandre Dias Tavares Costa, Dhélio Batista Pereira.

**Writing – original draft:** Rebecca K. Green, Emily Gerth-Guyette.

**Writing – review & editing:** Camilo Manchola, Michelle Oliveira Silva, Luiza Bastos Gottin, Kimberly E. Green, Alexandre Dias Tavares Costa.

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
