## [Decision Letter · Decision Letter 0]

20 Jul 2023

PONE-D-23-05526Integration of serial self-testing for COVID-19 as part of contact tracing in the Brazilian public health system: A pragmatic trial protocolPLOS ONE

Dear Dr. Green,

Thank you for submitting your manuscript to PLOS ONE. After careful consideration, we feel that it has merit but does not fully meet PLOS ONE’s publication criteria as it currently stands. Therefore, we invite you to submit a revised version of the manuscript that addresses the points raised during the review process.

ACADEMIC EDITOR:The protocol is relevant and will certainly simplify COVID-19 surveillance activities especially in low resource settings where access to highly sophisticated PCR tests are rarely available.Please address all comments provided by both reviewers. For reviewer 1, try to adjust your protocol's rational and methodology by making them more explicit to avoid any potential ambiguity. The revision should be minimal. 

Please also ensure that your revised submission fully complies with PLOS ONE publication criteria.

A rebuttal letter that responds to each point raised by the academic editor and reviewers. You should upload this letter as a separate file labeled 'Response to Reviewers'.A marked-up copy of your manuscript that highlights changes made to the original version. You should upload this as a separate file labeled 'Revised Manuscript with Track Changes'.An unmarked version of your revised paper without tracked changes. You should upload this as a separate file labeled 'Manuscript'.If applicable, we recommend that you deposit your laboratory protocols in protocols.io to enhance the reproducibility of your results. Protocols.io assigns your protocol its own identifier (DOI) so that it can be cited independently in the future. For instructions see: https://journals.plos.org/plosone/s/submission-guidelines#loc-laboratory-protocols. Additionally, PLOS ONE offers an option for publishing peer-reviewed Lab Protocol articles, which describe protocols hosted on protocols.io. Read more information on sharing protocols at https://plos.org/protocols?utm_medium=editorial-email&utm_source=authorletters&utm_campaign=protocols.

We look forward to receiving your revised manuscript.

Kind regards,

Ibrahim Jahun, MD, MSC, PhD

Academic Editor

PLOS ONE

Reviewers' comments:

Reviewer's Responses to Questions

**Comments to the Author**

1. Does the manuscript provide a valid rationale for the proposed study, with clearly identified and justified research questions?

Reviewer #1: Partly

Reviewer #2: Yes

2. Is the protocol technically sound and planned in a manner that will lead to a meaningful outcome and allow testing the stated hypotheses?

Reviewer #1: No

Reviewer #2: Yes

3. Is the methodology feasible and described in sufficient detail to allow the work to be replicable?

Reviewer #1: No

Reviewer #2: Yes

4. Have the authors described where all data underlying the findings will be made available when the study is complete?

Reviewer #1: Yes

Reviewer #2: Yes

5. Is the manuscript presented in an intelligible fashion and written in standard English?

Reviewer #1: Yes

Reviewer #2: Yes

6. Review Comments to the Author

You may also provide optional suggestions and comments to authors that they might find helpful in planning their study.

Reviewer #1: This article does not say anything about Co-Infections to COVID-19. Inclusion criteria should be elaborated with appropriate controls like, gold standard test controls to rule out possible chances of cross reactivity/ co-infectivity. If there are any cases of co-infections along with COVID-19 then whether this participant will be excluded or any other statistical methodology to analyze these populations? This criterion can increase the sensitivity/ Specificity of screening test.

Reviewer #2: 1- Study populations (line 108) Consideration for age range of 7 and above: Although the protocol has received ethical reviews and approvals, it might be necessary to consider limiting the study to adolescents, youths, and adults (aged 14 and above) this will limit the need for supervision of sample collection and self-testing processes by household members at home, to minimize potentials for spread of infections at household level during the process as part of infection prevention. Alternatively, the protocol has to detail how infection prevention will be enhanced through training and provision of PPEs to care givers who will supervise minors enrolled into this study, and what to do in case of exposure

2- Index cases (Line 109 – 110): indicate what method(s) of test was used or will be used for COVID-19 Positive index cases.

3- Randomization (line 119 – 121): the sentence indicated that contacts of index cases will be randomized, whereas randomization is only at the level of the index cases. This should be rephrased to show this. The use of randomization for the contacts of index cases throughout the rest of the protocol need to be rephrased accordingly.

4- Line 131: Will study participants receive the same brand and batch of self-test kits for consistency? This should be included in the narrative

5- Line 132: Include how the study participants will be contacted in this segment of the protocol – by physical visits, phone calls, WhatsApp (as indicated further down in the protocol), email or a mix

6- Line 149: Indicate if 3 results will be for 3 consecutive days or otherwise, and what is the rational for using 3 results out of 10-day daily testing?

7- Line 150: indicate if one-time Ag-RDT test of close contacts of index case within 2 days of symptoms onset, or within 7 days of positive results is the standard practice in Brazil – as proposed for the control group

8- While training of study participants is implied in the protocol, it would be necessary to include, as part of the methods, that training on standard sample collection, self-testing procedure and interpretation of results, as well as safety precautions and infection prevention and control methods, would be provided and by whom, and at what point of the study

9- Line 18 – 160: will the study participants be part of the focused group discussions – who are the stakeholders in this case? Will the FGD be conducted prior to enrollment into the study or at the completion of 10-day monitoring?

10- The protocol did not include any quality check or process for reviewing the Self-testing procedures and validating results. A quality control measure is essential to avoid over reporting of positive test results or under reporting. This could be done by review of captured images of the test strips or other alternative methods.

7. PLOS authors have the option to publish the peer review history of their article (what does this mean?). If published, this will include your full peer review and any attached files.

Reviewer #1: **Yes: **Manoj A R

Reviewer #2: **Yes: **McPaul I.J. Okoye

---

## [Author Response · Author response to Decision Letter 0]

21 Aug 2023

Dear Dr. Jahun,

Thank you for the review of this manuscript. The authors have carefully reviewed each comment made by yourself and the reviewers and have revised the text accordingly. Please see below for a point-by-point response to each comment.

1. The protocol is relevant and will certainly simplify COVID-19 surveillance activities especially in low resource settings where access to highly sophisticated PCR tests are rarely available. Please address all comments provided by both reviewers. For reviewer 1, try to adjust your protocol's rational and methodology by making them more explicit to avoid any potential ambiguity. The revision should be minimal. 

Response: Thank you for your comments. We have made minor edits to some of the methodological descriptions and rationale to help clarify them for the reader.

2. This article does not say anything about Co-Infections to COVID-19. Inclusion criteria should be elaborated with appropriate controls like, gold standard test controls to rule out possible chances of cross reactivity/ co-infectivity. If there are any cases of co-infections along with COVID-19 then whether this participant will be excluded or any other statistical methodology to analyze these populations? This criterion can increase the sensitivity/ Specificity of screening test. 

Response: The authors appreciate this attention to detail with regards to screening test performance. The objective of this study is effectiveness and operational feasibility of a novel service delivery mechanism using an approved product with well-documented and accepted test performance, including cross-reactivity among other respiratory pathogens. Because the focus of this study was not diagnostic performance, but rather effectiveness of the service delivery, the authors sought to keep eligibility criteria as pragmatic and reflective of the real-world use case as possible. The study utilized the GeneFinder™ COVID-19 Ag-RDT by Osang Healthcare, which is approved for commercial use and sale in Brazil. According to the product insert, when tested in triplicate with several other human coronaviruses, adenoviruses, parainfluenza viruses, influenza A & B, enterovirus, RSV A & B,

rhinovirus, MERS, streptococcus pneumoniae, legionella pneumophila, candida albicans, streptococcus pyogenes, mycoplasma pneumoniae, and haemophilus influenzae, no cross-reactivity or interference was found.

3. Study populations (line 108): Consideration for age range of 7 and above: Although the protocol has received ethical reviews and approvals, it might be necessary to consider limiting the study to adolescents, youths, and adults (aged 14 and above) this will limit the need for supervision of sample collection and self-testing processes by household members at home, to minimize potentials for spread of infections at household level during the process as part of infection prevention. Alternatively, the protocol has to detail how infection prevention will be enhanced through training

and provision of PPEs to care givers who will supervise minors enrolled into this study, and what to do in case of exposure 

Response: The study team thoroughly considered the acceptable age range for inclusion during the study design phase and throughout the approval process. The ST used in this study is indicated for use by individuals aged 14 or older who self-collect their swabs as well as in children aged 2 or older where an adult collects the swab. The study team felt that excluding school-aged youth under 14 years old would neglect a crucial study population who could benefit from this type of novel service delivery model. Testing procedures in this study were implemented according to the manufacturer’s instructions for use, which describe basic hygienic steps to be taken by anyone performing the test.

4. Index cases (Line 109 – 110): indicate what method(s) of test was used or will be used for COVID-19 Positive index cases.

Response: Additional clarification has been added to line 127 to the extent possible. Index cases were tested per local testing practices and referred to the study team for enrollment; the conduct of screening tests to identify index cases was not under research purview.

5. Randomization (line 119 – 121): the sentence indicated that contacts of index cases will be randomized, whereas randomization is only at the level of the index cases. This should be rephrased to show this. The use of randomization for the contacts of index cases throughout the rest of the protocol need to be rephrased accordingly.

Response: Thank you for flagging this ambiguity; the wording has been revised throughout.

6. Line 131: Will study participants receive the same brand and batch of self-test kits for consistency? This should be included in the narrative 

Response: All participants received the same brand and lot of self-test kits. This has been added to line 151.

7. Line 132: Include how the study participants will be contacted in this segment of the protocol – by physical visits, phone calls, WhatsApp (as indicated further down in the protocol), email or a mix 

Response: Line 146 has been updated to explicitly state that enrollment visits occurred in person, as this was necessary to complete the professional test.

8. Line 149: Indicate if 3 results will be for 3 consecutive days or otherwise, and what is the rational for using 3 results out of 10-day daily testing?

Response: Clarification has been added to line 183 indicating the three results may be consecutive or otherwise. Despite best efforts to ensure 100% compliance of follow-up, the study team recognized that the likelihood of achieving 100% follow-up survey completion was highly unlikely. Given the likelihood of a persistent positive test for participants who converted to positive, the study team reasoned it would be likely for a positive result to be captured if at least three results were documented.

9. Line 150: indicate if one-time Ag-RDT test of close contacts of index case within 2 days of symptoms onset, or within 7 days of positive results is the standard practice in Brazil – as proposed for the control group

Response: As we have observed throughout the pandemic, COVID-19 practices are highly variable even within a single country or region and evolve over time. At the time of protocol development and study start, practices for contact tracing varied widely across the country depending on health facility bandwidth and local disease epidemiology. As such, the study team based eligibility on a wide range of input, including literature on COVID-19 transmission windows and current-at-the-time Brazilian guidelines for isolation of individuals exposed to someone with COVID-19, recognizing that testing of exposed individuals was being done inconsistently across the country.

10. While training of study participants is implied in the protocol, it would be necessary to include, as part of the methods, that training on standard sample collection, self-testing procedure and interpretation of results, as well as safety precautions and infection prevention and control methods, would be provided and by whom, and at what point of the study

Response: As a pragmatic trial, one of the objectives of this study was to evaluate the extent to which formally untrained people are able to correctly perform a ST given standard manufacturer-provided materials. Study participants were not meant to be formally trained on self-test use by a health professional because this is not what would happen in the real world should a person choose to purchase a ST for their own use. Rather, the health professional observed participants perform the self-test during their enrollment visit, after the participant reviewed the manufacturer-provided training materials, and only intervened if the participant made a critical error.

11. Line 18 – 160: will the study participants be part of the focused group discussions – who are the stakeholders in this case? Will the FGD be conducted prior to enrollment into the study or at the completion of 10-day monitoring?

Response: Stakeholders are inclusive of study participants and other health professionals; this has been clarified in line 194. As the FGDs are meant to learn about acceptability and preferences for self-testing, they cannot be completed until the participants have completed their study follow-up period.

12. The protocol did not include any quality check or process for reviewing the Self-testing procedures and validating results. A quality control measure is essential to avoid over reporting of positive test results or under reporting. This could be done by review of captured images of the test strips or other alternative methods.

Response: The quality check for participant performance of the self-testing procedure was the supervised ST conducted during the enrollment visit. If the participant performed the test correctly with no critical errors, they were considered proficient in ST use for the study. Participants were asked to upload a picture of their ST result each day they performed one, but due to the variable nature of photos and associated challenges with accurately reading RDT results from a photo, the study team felt it would be most appropriate for the participant-reported result to remain the ground truth.

We hope that the revised manuscript will merit publication and appreciate your reconsideration of this important work.

Sincerely,

Rebecca Green, on behalf of the authors

---

## [Decision Letter · Decision Letter 1]

18 Sep 2023

Integration of serial self-testing for COVID-19 as part of contact tracing in the Brazilian public health system: A pragmatic trial protocol

PONE-D-23-05526R1

Dear Dr. Green,

We’re pleased to inform you that your manuscript has been judged scientifically suitable for publication and will be formally accepted for publication once it meets all outstanding technical requirements.

Kind regards,

Ibrahim Jahun, MD, MSC, PhD

Academic Editor

PLOS ONE

Additional Editor Comments (optional):

Reviewers' comments:

Reviewer's Responses to Questions

**Comments to the Author**

1. Does the manuscript provide a valid rationale for the proposed study, with clearly identified and justified research questions?

Reviewer #2: Yes

2. Is the protocol technically sound and planned in a manner that will lead to a meaningful outcome and allow testing the stated hypotheses?

Reviewer #2: Yes

3. Is the methodology feasible and described in sufficient detail to allow the work to be replicable?

Reviewer #2: Yes

4. Have the authors described where all data underlying the findings will be made available when the study is complete?

Reviewer #2: Yes

5. Is the manuscript presented in an intelligible fashion and written in standard English?

Reviewer #2: Yes

6. Review Comments to the Author

You may also provide optional suggestions and comments to authors that they might find helpful in planning their study.

Reviewer #2: Authors have addressed the issues and observations made in the first iteration, and have made further improvements to the manuscript.

The following minor improvements are further recommended:

1- It would be necessary to include measures the authors would implement to minimize potentials for viral transmission to household members (infection prevention and control measures) at the household level during self testing for all study participants in the the intervention arm, especially for children requiring supervision by household members. This could be included in the safety or ethical consideration sections

2- In figure 2 and Ethical Consideration: Authors to clarify what decision would be made if the Professional Ag.RDT conducted at the enrollment stage was positive - will study participants in the control arm who tested positive be linked to care? This should be included in the ethical consideration section.

3- Authors to clarify what decision would be made for study participants in the Intervention ARM, who had a positive result based on Professional Ag.RDT at enrolment stage, but negative results following 10 days of serial Self-Testing (ST) without any significant testing procedure error, would these category of study participants be linked to care as per Brazil's standard guidelines?

4- Authors may wish to consider including study participants inclusion and exclusion criteria sections in the protocol to facilitate understanding

7. PLOS authors have the option to publish the peer review history of their article (what does this mean?). If published, this will include your full peer review and any attached files.

Reviewer #2: **Yes: **McPaul I.J. Okoye

---

## [Editor Report · Acceptance letter]

25 Sep 2023

PONE-D-23-05526R1 

Integration of serial self-testing for COVID-19 as part of contact tracing in the Brazilian public health system: A pragmatic trial protocol 

Dear Dr. Green:

I'm pleased to inform you that your manuscript has been deemed suitable for publication in PLOS ONE. Congratulations! Your manuscript is now with our production department. 

Kind regards, 

on behalf of

Dr. Ibrahim Jahun 

Academic Editor

PLOS ONE